# The Preclinical Evaluation of a Second-Generation Antivenom for Treating Snake Envenoming in India

**DOI:** 10.3390/toxins14030168

**Published:** 2022-02-24

**Authors:** Saurabh Attarde, Ashwin Iyer, Suyog Khochare, Umesh Shaligram, Mayur Vikharankar, Kartik Sunagar

**Affiliations:** 1Evolutionary Venomics Lab, Centre for Ecological Sciences, Indian Institute of Science, Bangalore 560012, Karnataka, India; asaurabh@iisc.ac.in (S.A.); iyer.ash.17@gmail.com (A.I.); suyogk@iisc.ac.in (S.K.); 2Serum Institute of India Pvt. Ltd., 212/2, Hadapsar, Off Soli Poonawalla Road, Pune 411028, Maharashtra, India; umesh.shaligram@seruminstitute.com (U.S.); mayur.vikharankar@seruminstitute.com (M.V.)

**Keywords:** snakebite, antivenom therapy, second-generation antivenom, ‘big four’ snakes

## Abstract

Snake envenoming afflicts the Indian subcontinent with the highest rates of mortality (47,000) and morbidity globally. The only effective treatment for snakebites is the administration of antivenom, which is produced by the hyperimmunisation of equines. Commercial Indian antivenoms, however, have been shown to exhibit a poor preclinical performance in neutralising venom, as a result of inter- and intrapopulation snake venom variation. Additionally, their poor dose effectiveness necessitates the administration of larger volumes of antivenom for treatment, leading to several harmful side effects in snakebite victims, including serum sickness and fatal anaphylaxis. In this study, we employed chromatographic purification to enhance the dose efficacy of commercial Indian antivenoms. The efficacy of this ‘second-generation’ antivenom was comparatively evaluated against six other marketed antivenoms using a number of in vitro and in vivo preclinical assays, which revealed its superior venom recognition capability. Enhanced purity also resulted in significant improvements in dose effectiveness, as the ‘second-generation’ antivenom exhibited a 3 to 4.5 times increased venom neutralisation potential. Furthermore, preclinical assays revealed the increased effectiveness of the ‘second-generation’ antivenom in countering morbid effects inflicted by the ‘big four’ Indian snakes. Thus, we demonstrate the role of simpler purification steps in significantly enhancing the effectiveness of snakebite therapy in regions that are most affected by snakebites.

## 1. Introduction

The World Health Organization (WHO) recognises snake envenoming, a socioeconomic disease that has plagued the Indian subcontinent (which experiences the highest rates of mortality and morbidity due to this issue globally), as a priority neglected tropical disease [1]. Conventional antivenoms that are manufactured by hyperimmunising equines are the only effective treatment for snakebites [2]. However, commercial Indian antivenoms have been documented to exhibit a poor preclinical efficacy in neutralising venom due to inter- and intrapopulation venom variation in targeted species [3,4,5,6,7], as well as closely related and medically important yet neglected snakes (a.k.a., the ‘neglected many’) [8,9,10,11,12,13]. They are also known to exhibit a poor dose efficacy, resulting in the need for larger volumes of antivenom doses in order to effect a cure. This, perhaps, frequently results in severe allergic reactions, including serum sickness and fatal anaphylaxis in patients receiving these antivenoms [14]. Recombinant antivenoms with increased potency, paraspecificity, and cost-effectiveness are touted as alternative solutions for addressing these major shortcomings of conventional antivenoms. Unfortunately, however, these antivenoms are realistically several years to at least a decade away from being available for the treatment of snakebite victims. Hence, in addition to exploring recombinant technology for the production of effective snakebite therapeutics, there is an urgent need to improve the efficacy of existing antivenom products.

In this study, by employing the chromatographic purification of the bulk, we significantly enhanced the preclinical performance of conventional Indian antivenom products. The effectiveness of test batches of this ‘second-generation’ antivenom was evaluated using a variety of in vitro and in vivo preclinical assays, including venom recognition and toxicity and pathology neutralisation. The outcomes of these experiments demonstrated the significantly superior performance of the purified product over all other major commercial Indian antivenoms. Thus, we demonstrate the feasibility and effectiveness of employing simpler purification and processing steps for the immediate improvement of existing antivenom products.

## 2. Results

### 2.1. Physicochemical Properties

The physicochemical properties of the commercial Indian and second-generation SIIPL antivenoms, which were manufactured using the ‘big four’ snake venoms from Tamil Nadu (Haffkine being the exception, as it was manufactured using snake venoms in Maharashtra), are listed in Table 1. While a reducing SDS-PAGE confirmed the formulation of the antibodies present in the antivenom (Figure 1), other parameters were manually evaluated. The antivenoms differed in their physical appearance before and after reconstitution, odour, pH, turbidity, and the time required for complete reconstitution in physiological saline (Table 1).

### 2.2. SDS-PAGE Profiling of Antivenoms

The three batches of second-generation antivenom, when subjected to 15% reducing SDS-PAGE, exhibited nearly identical profiles to their conventional antivenom counterparts, suggesting that they contained an F(ab’)_2_ formulation. Electrophoresis revealed the presence of two bands between 25 and 37 kDa (Figure 1), which correspond to light and heavy chains of F(ab’)_2_ molecules, respectively, consistent with the previously reported profiles of F(ab’)_2_ preparations of equine origin [15,16,17,18,19,20].

### 2.3. Size-Exclusion HPLC (SE-HPLC)

SE-HPLC was employed for assessing the purity of F(ab’)_2_ in conventional Indian antivenoms and the three batches of second-generation antivenom. The latter showed the highest purity, with profiles characterised by a single symmetrical peak (Figure 2). On the other hand, the peak tailing in the profiles of the conventional antivenoms indicated the presence of impurities (Figure 2). These impurities were later identified as other plasma proteins, such as albumin, alpha-macroglobulin, fibrinogen, fibronectin, haptoglobin, plasminogen, prothrombin, and serpin [19]. This demonstrates the increased purity of the second-generation antivenom compared to its conventional counterparts. We also detected a minor peak of F(ab’)_2_ dimers [19] at a 10 min retention time in the third batch of the second-generation antivenom (SIIPL-03) and the antivenom manufactured by Biological E.

### 2.4. In Vitro Venom-Recognition

The ‘big four’ snake venom-recognition potential of conventional antivenoms and the three batches of the second-generation antivenom were assessed using quick-screening ELISAs and immunoblotting experiments. In the quick-screening ELISAs, only three dilutions of the antivenom were incubated with a fixed venom concentration, and the absorbance values at 405 nm, which directly correlate to the amount of antivenom antibody-venom protein binding, were plotted (Figure 3). These dilutions were chosen based on the preliminary standard ELISA testing of one of the antivenom samples against the ‘big four’ venoms. The titre dilution, along with two dilutions around this value, were chosen for the experiment. The IgG from unimmunised horses was used as a negative control.

In these experiments, the second-generation antivenom exhibited the highest binding against elapid venoms (titre: 1:500; *N. naja*: 2.84 Optical Density (OD); *B. caeruleus*: 1.86 OD) among all of the tested antivenoms (*p* ≤ 0.01), while the conventional antivenom manufactured by Virchow bound slightly better (*p* ≤ 0.01) to the viperid venoms (titre: 1:500; *D. russelii*: 2.53 OD; *E. carinatus*: 2.67 OD; Figure 3). In contrast, the antivenoms manufactured by Bharat Serums and Haffkine poorly recognised the ‘big four’ snake venoms. The Premium Serums antivenom (titre: 1:500; *N. naja*: 1.28 OD; *D. russelii*: 1.35 OD; *B. caeruleus*: 0.75 OD; *E. carinatus*: 1.24 OD) exhibited the least binding among all the tested antivenoms against the ‘big four’ snake venoms (*p* ≤ 0.01). Overall, the three batches of the second-generation antivenom exhibited an increased recognition of ‘big four’ venoms, binding relatively better than all the other tested conventional antivenoms.

Additionally, immunoblotting experiments were performed to elucidate the ability of antivenoms to recognise different toxins in the ‘big four’ snake venoms. The second-generation antivenoms, along with the VINS and Virchow antivenoms, were found to recognise many low-, mid-, and high-molecular-weight toxins present in these venoms (Appendix A). The antivenoms manufactured by Premium Serums and Biological E. showed an intermediary binding to various venom toxins, whereas the Haffkine and Bharat Serums antivenoms poorly recognised the ‘big four’ venoms. Interestingly, the Haffkine antivenom failed to bind to the majority of toxins present in *E. carinatus* venom, revealing the inadequate venom-binding capabilities of this antivenom against the ‘big four’ venoms, corroborating the findings of the ELISA experiments (Figure 3).

The results of the in vitro binding experiments revealed the better venom recognition capability of the second-generation antivenoms, particularly SIIPL-01, against the ‘big four’ snake venoms. The improved in vitro binding efficiency of the second-generation antivenoms could result from multiple refinements, such as a higher proportion of toxin-binding antibodies, increased purity, or a combination thereof. Among the conventional antivenoms, the antivenom manufactured by Virchow was found to perform worst.

### 2.5. Immunochromatography

Immunochromatography experiments revealed the venom binding strengths of antivenoms and the identity of toxins that they recognised and failed to recognise. The immunoreactivities of the second-generation antivenoms (SIIPL-01) and two conventional antivenoms (Virchow and Premium Serums) against the venoms of the ‘big four’ snakes were further investigated by immunochromatography. The two antivenoms were selected based on the outcomes of ELISA experiments, with the SIIPL-01 and Virchow antivenoms exhibiting the highest binding and the Premium Serums antivenom binding relatively poorly. The examination of the RP-HPLC profiles of the whole venom and the retained (venom components recognised by the antivenom) and non-retained (venom components that are not recognised by the antivenom) fractions revealed that the second-generation antivenom (SIIPL-01) exhibited the highest binding to the majority of venom proteins and that very few components were found in the non-retained fraction (Figure 4). The Virchow antivenom was identified as exhibiting the second-best binding after SIIPL-01, as the majority of venom proteins were found in the retained fraction. However, the Virchow antivenom demonstrated a poorer immunological recognition against the *B. caeruleus* venom (Figure 4). Further, the Premium Serums antivenom was found to be a relatively worse performer among the tested antivenoms, as most venom components were observed in the non-retained fraction (Figure 4). These findings were in line with the outcomes of ELISA and Western blotting experiments.

### 2.6. Mass Spectrometric Analyses of Antivenoms

To evaluate the proportion of immunoglobulins and impurities, three antivenoms were selected for mass spectrometry (LC-MS/MS) experiments. These antivenoms were selected on the basis of their in vitro venom recognition potential. LC-MS/MS analysis revealed that IgGs constituted the majority of the protein content, while contaminants (e.g., albumin, alpha-macroglobulin, fibrinogen, fibronectin, haptoglobin, plasminogen, prothrombin, and serpin) formed a minor fraction of all three antivenoms (Appendix A).

### 2.7. Venom Neutralisation Potential of Antivenoms

The toxicity profiles (or LD_50_) of the ‘big four’ snake venoms were determined before conducting the median effective dose (ED_50_) experiments (Figure 5). Given the ethical considerations, only a single best binding conventional antivenom (Virchow) was selected to assess the neutralisation potential compared to the best binding second-generation antivenom (SIIPL-01). The results of these in vivo experiments revealed that the second-generation antivenom (SIIPL-01) exhibited a significant improvement in terms of its neutralisation potential against the Elapidae (*N. naja*: 2.74 mg/mL and *B. caeruleus:* 2.08 mg/mL) and Viperidae (*D. russelii*: 1.91 mg/mL and *E. carinatus*: 1.49 mg/mL) venoms. In comparison to the marketed potencies of conventional antivenoms (*N. naja* and *D. russelii*: 0.60 mg/mL; *B. caeruleus* and *E. carinatus*: 0.45 mg/mL), SIIPL-01 exhibited a 3.1 to 4.6 times improvement in its neutralisation potency. Similarly, the conventional antivenom produced by Virchow surprisingly exhibited a neutralisation potency that was between 1.4 and 1.5 times better (*N. naja*: 0.951 mg/mL and *B. caeruleus*: 0.636 mg/mL) than the marketed values against these elapid snakes (Figure 5B). However, the neutralisation potential of this antivenom against *E. carinatus* venom (2.44 mg/mL) was found to be 5.4 times that of the marketed value of commercial antivenoms and 1.6 times greater than the second-generation antivenom (Figure 5B).

### 2.8. Morbidity Assays

The significant improvement in the venom neutralisation capabilities of the second-generation (SIIPL-01) antivenom and the conventional Virchow antivenom, consistent with their superior in vitro performance, is promising. While both antivenoms were considerably effective in countering the mortalities inflicted by the ‘big four’ snakes, it is imperative to understand their ability to neutralise snake venom-induced morbidity, particularly with the varying pathologies that result from viperid snake envenoming. Therefore, we conducted WHO-recommended preclinical assays to evaluate the abilities of these antivenoms in countering pathologies, such as haemorrhage and necrosis.

The haemorrhagic potential of viper venoms was directly proportional to the concentration of the venom (Appendix A). The minimum haemorrhagic dose (MHD) for *E. carinatus* venom (0.44 µg/mouse) from southern India was five times lower in comparison to that of *D. russelii* (2.19 µg/mouse), highlighting the extreme haemorrhagic nature of the saw-scaled viper venom (Figure 6A,D, Appendix A). Following the estimation of the MHD of viper venoms, the haemorrhage neutralisation capabilities of the antivenoms were evaluated by determining their MHD-ED_50_ values. The second-generation antivenom (SIIPL-01) was found to exhibit 2.1 times the haemorrhage neutralisation potential against the *D. russelii* venom (MHD-ED_50_: 30.20 mg/mL) than the conventional Virchow antivenom (MHD-ED_50_: 14.10 mg/mL; Figure 6B,C and Appendix A). Similarly, SIIPL-01 also exhibited a better neutralisation of the *E. carinatus* venom (MHD-ED_50_: 2.77 mg/mL) than the conventional product manufactured by Virchow (MHD-ED_50_: 2.23 mg/mL; Figure 6E,F and Appendix A).

To determine the necrotic abilities of viper venoms from Southern India, the WHO recommended that MND studies be conducted. Here, distinct concentrations of viperid venoms were injected intradermally and the necrotic symptoms on the skin were noted 72 h post-venom injections. Mice injected with the *E. carinatus* venom from Southern India (15–40 µg) exhibited signs of hemorrhage, but did not show necrosis (Appendix A). Similarly, mice injected with the *D. russelli* venom exhibited mild haemorrhagic symptoms, but did not show any signs of necrosis (Appendix A).

## 3. Discussion

### 3.1. Purification Steps Can Significantly Enhance the Dose-Effectiveness of Commercial Indian Antivenoms

The proportion of venom-specific neutralising antibodies in antivenoms has an obvious impact on their efficacy and potency. The greater the proportion of neutralising antibodies directed towards medically important toxins is, the more effective and potent the antivenom will be in neutralising snake envenomation. The toxin-targeting antibody content of an antivenom can be enhanced by various means, including chromatographic purification and affinity chromatography [21,22]. However, since these purification steps are not mandated by the WHO for the commercial production of antivenoms [19], antivenom manufacturers tend to avoid such steps to reduce production costs.

The purification steps implemented in the production of the second-generation antivenom in this study revealed the impact of preprocessing steps, as these resulted in a significant improvement in the overall quality of the product. The second-generation antivenoms exhibited clean SE-HPLC profiles in comparison to their conventional counterparts, highlighting the significant improvements made in purity (Figure 2). Similarly, the second-generation antivenom contained a single symmetrical peak in size-exclusion chromatography experiments (Figure 2), whereas the tailing of the peaks was observed in the profiles of the conventional antivenoms (Figure 2). Thus, these findings highlight the importance of utilising simpler chromatographic purification steps for the manufacture of commercial antivenoms. Conventional Indian antivenoms, which have a lower proportion of venom toxin-binding and neutralising antibodies [3,4,8,10,12,13] and a higher load of impurities, often result in severe secondary reactions to the antivenom, including serum sickness and fatal anaphylaxis [14]. Hence, the improvements in the purity and overall quality of the antivenom product achieved here are anticipated to greatly reduce such undesirable effects. However, it should be noted that clinical studies are warranted to elucidate the precise role of purity in reducing undesirable secondary reactions to antivenoms.

### 3.2. The Purified Second-Generation Antivenom Exhibits Enhanced In Vitro Venom Recognition and In Vivo Venom Neutralisation Potential Compared to Its Conventional Antivenom Counterparts

In addition to being relatively more pure in comparison to conventional antivenom products, the second-generation antivenoms were also found to exhibit enhanced in vitro venom recognition and in vivo venom neutralisation potential. The tested SIIPL-01 antivenom was characterised by a 3 to 4.5 times increased toxicity neutralisation potential compared to the commercially marketed values of neutralisation against the ‘big four’ snake venoms. Since most Indian commercial antivenoms exhibit much lower preclinical neutralisation potencies than these marketed values [3,4,8,10,11], these improvements are substantial. Surprisingly, unlike the preclinical performance of most conventional antivenoms, the Virchow antivenom product exhibited an enhanced neutralisation of venom-induced lethality in a mouse model of envenoming. In fact, its neutralisation potency surpassed that of the second-generation antivenom when tested against the venom of *E. carinatus*.

### 3.3. The Second-Generation Antivenom also Exhibits Significant Improvements in the Morbidity Neutralisation Potential

A variety of morbid effects are often associated with snake envenoming. The ‘big four’ snakes, particularly the viper species, are infamous for inflicting morbidity in a large number of Indians annually. Therefore, while both the second-generation antivenom and the conventional Virchow antivenom exhibited superior venom recognition and venom-induced lethality neutralisation potential, they are also expected to neutralise snakebite-inflicted pathologies, including haemorrhage, necrosis, and myotoxicity. In our WHO-recommended preclinical assays, SIIPL-01 antivenom exhibited twice the neutralisation potential as the conventional antivenom against the hemorrhagic symptoms induced by the Russell’s viper venom (Figure 6B,C). While the conventional Virchow antivenom was able to neutralise the lethal effects inflicted by *E. carinatus* venom more effectively than SIIPL-01, it exhibited a relatively lower performance than the latter in countering the significant haemorrhagic effects of this snake venom (Figure 6E,F). Mice injected with either *E. carinatus* or *D. russelii* venoms from Southern India did not exhibit necrotic effects. These findings highlight the superior performance of the second-generation antivenom in countering the morbid effects inflicted by the ‘big four’ Indian snakes.

### 3.4. The ‘Perfect’ Antidote: Formulation of India’s Next-Generation Antivenom Product

Recent studies have unravelled the inefficaciousness of Indian antivenoms in countering the pathologies inflicted by the disparate populations of the ‘big four’ snakes and their close relatives [3,4,6,7,8,9,11,12,13]. Not only were the antivenoms ineffective against medically important snake species and their populations across very large distances, but they were also found to be inadequate in neutralising the intrapopulation venom variation at much finer geographical scales [10]. The inefficaciousness of Indian antivenoms stems from the geographically restricted sources of venoms used in the antivenom manufacturing process. The immunisation mixture used in this process consists of venoms sourced from the ‘big four’ snakes in the southernmost part of the country and the resultant product is used for snakebite treatment across India. Therefore, this polyvalent antivenom fails to counter the tremendous intra- and interspecific venom variations documented in medically important Indian snakes. Moreover, despite this inferior preclinical performance, Indian antivenoms are also used for the treatment of envenomings by very distant populations of the ‘big four’ snakes in neighbouring regions, including Pakistan, Sri Lanka, and Bangladesh [23,24,25]. While the second-generation antivenom evaluated in this study exhibits appreciable benefits in countering venom-induced toxicity and morbidity, similarly to its conventional antivenom counterparts, it is unlikely to exhibit pan-India effectiveness and paraspecificity, as it still relies on this century-old technology. Therefore, in addition to the improvements made in the purity of the antivenom product, there is a pressing need to refine the formulation of the immunisation mixture used in the manufacturing process. This can be achieved through the identification of medically important snakes by region (i.e., both ‘big four’ and the ‘neglected many’) and the inclusion of their venoms in the immunisation mixture for the production of regionally effective antivenoms. Such a regional antivenom product, manufactured by incorporating purification processes, is not only expected to reduce the incidence of undesirable secondary reactions of the antiserum therapy, but will also offer pan-India effectiveness for the treatment of snakebites.

### 3.5. Limitations of the Study

Given the ethical and financial constraints of this study, a single batch of the respective Indian antivenoms and three test batches of the ‘second-generation’ antivenom were investigated in this study. Therefore, rigorous statistical comparisons and clinical research are necessary to validate the superior performance and commercialisation of the ‘second generation’ antivenom.

## 4. Conclusions

The polyvalent antivenom produced by hyperimmunising equines with the ‘big four’ Indian snake venoms is the only scientifically approved treatment for snakebites. In fact, this is the only antivenom marketed throughout the country, and its strategy of production has remained unchanged since its inception over a hundred years ago. Recent studies have documented the inefficacy of these antivenoms against geographically disparate populations of the ‘big four’ snakes, as well as against other medically important yet neglected species [3,4,8,9,10,11,12,13]. These inadequacies underscore the pressing need to develop next-generation recombinant antivenoms with increased potency, paraspecificity, and cost-effectiveness. Unfortunately, however, these products are far from fruition, as they are several years away from being commercially available to snakebite victims in regions affected by snakebites. Therefore, it is imperative to enhance the efficacies of currently marketed antivenom products. The second-generation antivenom produced and preclinically evaluated in this study promises to be one such step in this direction. We demonstrate that the improvements made in the production of the existing antivenom manufacturing process can also lead to commensurate improvements in the dose effectiveness and neutralising potencies of commercial antivenoms. Our findings call for the streamlining of antivenom manufacturing protocols amongst the leading antivenom producers in the country. These improvements are anticipated to save the lives and limbs of hundreds and thousands of Indian snakebite victims annually.

## 5. Materials and Methods

### 5.1. Snake Venoms and Antivenoms

The ‘big four’ snake venoms were procured from the Irula Snake Catchers’ Industrial Cooperative Society (ISCICS), Tamil Nadu, and stored at −80 °C until use. Here, three batches (SIIPL-01, SIIPL-02, and SIIPL-03) of the ‘second-generation’ antivenom (10 L) were prepared by hyperimmunising healthy equines with the ‘big four’ snake venoms from Tamil Nadu using the optimised protocol described in the WHO Guidelines for the production, control, and regulation of snake antivenom immunoglobulins [26,27]. The hyperimmune plasma was diluted with sterile water for injection (SWFI) in a 1:2 proportion and subjected to pepsin digestion for 4 h at 37 °C. IgG molecules in the plasma were digested using 0.1% pepsin (Thermo Fisher Scientific, Waltham, MA, USA) at pH 3.5 with continuous stirring at 450 RPM. The digestion process was quenched by increasing the pH to 6.0 and then heated at 58 °C for thermocoagulation, which leads to the precipitation and denaturation of thermolabile proteins. Thermocoagulation was followed by caprylic acid precipitation at a final concentration of 3.0%. The reaction mixture was vigorously stirred at 650 RPM to eliminate non-IgG molecules (e.g., albumin, globulin, etc.). The resultant mixture was then centrifuged at 200 RPM to remove semi-solid slurry containing small peptides, high-molecular-weight aggregates, pepsin residues, caprylic acid and traces of albumin. Following centrifugation, the supernatant, containing F(ab’)_2_ fragments and traces of aforementioned impurities, was subjected to chromatographic purification.

The purified antivenom product (henceforth, referred to as the ‘second-generation’ antivenom) was manufactured by Serum Institute of India Pvt. Ltd. (SIIPL), Pune, using the chromatographic techniques outlined below. Additional descriptions of the investigated second-generation antivenom and its conventional counterparts—namely, Premium Serums, VINS BioProducts, Bharat Serums, Haffkine Pharmaceuticals, Virchow Biotech and Biological E.—are provided in Table 1.

### 5.2. Chromatographic Purification

During the purification process, the supernatant solution containing desired F(ab’)_2_ antibody fragments and impurities (e.g., small peptides, high molecular weight aggregates, pepsin residues, caprylic acid, and trace amounts of albumin) was passed through a chromatography column containing an ion-exchange resin (Bio-Rad Laboratories, Hercules, CA, USA) equilibrated with acetate buffer at a flow rate of 4.5 mL/min. Antibody molecules that exhibited an increased affinity towards the resin were retained, whereas the contaminant proteins were collected as they flowed through. After a residence time of 10 min, the desired antibody fractions were then eluted using an acetate buffer and salt gradient. The fractions were tested for protein content and purity level with Ultra-High-Performance Liquid Chromatography (UHPLC). The resultant purified fractions were pooled and subjected to Introduction to Tangential Flow Filtration using a 30 kDa polyethersulfone membrane. The concentrated product was then formulated with Glycine, NaCl, and cresol, before being filtered through a 0.2-micron filter to eliminate microbiological contamination. The finished product was aseptically transferred into sterile glass vial and subjected to lyophilisation.

### 5.3. Physicochemical Characterisation

Important physicochemical features of the antivenom, such as colour, odour, turbidity, pH and solubility, were evaluated following the WHO guidelines (Table 1; [26]). Briefly, vials of the conventional and the second-generation antivenom were inspected manually to examine the quality, appearance of lyophilised powder and the presence of moisture. The vials were then carefully reconstituted in sterile water and the time taken for complete reconstitution was recorded.

### 5.4. Protein Quantification

Bradford method was used to estimate the protein concentrations of antivenoms, with Bovine Serum Albumin (BSA) as the standard. The total IgG content was measured using the Bovine Gamma Globulin (BGG) standard curve after antivenom vials were reconstituted according to the manufacturer’s instructions (Table 1).

### 5.5. One Dimensional SDS-PAGE

Antivenoms (8 μg) were separated on a 15% sodium dodecyl sulphate-polyacrylamide gel electrophoresis (SDS-PAGE) under reducing conditions, then stained with Coomassie Brilliant Blue R-250 (Sisco Research Laboratories Pvt. Ltd., Mumbai, India) and visualized using an iBright CL1000 gel documentation system (Thermo Fisher Scientific, USA).

### 5.6. Size-Exclusion HPLC (SE-HPLC)

The quality of conventional Indian antivenoms and the three batches of second-generation antivenoms was assessed using SE-HPLC, with minor modifications to a previously reported method [28]. The SE-HPLC system was kept at 25 °C and consisted of a Shimadzu LC-20AD series HPLC system (Kyoto, Japan), a photodiode array detector (PDA), and a Bio-diol column (4.6 × 300 mm, 5 m particle size; Shimadzu, Kyoto, Japan). Proteins were eluted at a flow rate of 1 mL/min for 50 min using a 0.05 M sodium phosphate-0.15 M NaCl buffer (pH 7.0). PDA detector responses were collected at 280 nm with a sample injection volume of 20 μL.

### 5.7. Mass Spectrometry (LC-MS/MS)

Antivenom samples were further characterised by tandem mass spectrometry using a previously reported approach [9], in which the whole antivenom samples were treated with 10 mM of dithiothreitol (DTT), alkylated with 30 mM of iodoacetamide (IAA), and then digested overnight at 37 °C with trypsin (0.2 μg/μL). The digested antivenom samples were then put through a Thermo EASY nLC 1200 series system with a C18 nano-LC column (50 cm × 75 μm, 3 μm particle size, and 100 Å pore size) (Thermo Fisher Scientific, MA, USA). Buffer A (0.1% formic acid in HPLC grade water) and gradient buffer B (0.1% formic acid in 80% acetonitrile) were used in liquid chromatography for 120 min at a constant flow rate of 300 nL/min, with 10–45% over 98 min, 45–95% over 4 min, and 95% over 18 min. The antivenom samples were characterised using a Thermo Orbitrap Fusion Mass Spectrometer (Thermo Fisher Scientific, MA, USA). MS scans were performed with the following parameters: a scan range (*m*/*z*) of 375–1700, a resolution of 120,000, and a maximum injection duration of 50 ms. For fragment scans (MS/MS), an ion trap detector with high collision energy fragmentation (30%), a scan range (*m*/*z*) of 100–2000, and a maximum injection duration of 35 ms were used. Raw MS/MS spectra were searched against the National Center for Biotechnology Information’s (NCBI) non-redundant (nr) database (Equus caballus: 9796; September 2021) to identify antibodies and other proteins present in the antivenom samples using the PEAKS Studio X Plus software (Bioinformatics Solutions Inc., Waterloo, ON, Canada). The tolerances for parent and fragment mass errors were chosen at 10 ppm and 0.6 Da, respectively. Fixed modifications included cysteine carbamidomethylation, whereas variable modifications included methionine oxidation and asparagine or glutamine deamidation. A maximum of two missed cleavages by trypsin were allowed in the semispecific mode. FDR of 0.1%; minimum number of unique peptides: 1; and −10lgP protein score of 40 were the match acceptance filtering parameters. Mass spectrometry data were deposited in the ProteomeXchange Consortium via the PRIDE [29] partner repository with the data identifier: PXD029436.

### 5.8. Enzyme-Linked Immunosorbent Assay (ELISA)

The in vitro binding affinity of conventional Indian antivenoms and the three batches of the second-generation antivenom were evaluated using a previously described indirect ELISA experiment [8,30] with substantial modifications. Here, we employed a quick ELISA screening strategy, wherein specific dilutions of antivenoms were selected based on preliminary ELISA experiments. This altered strategy dramatically reduced the experimentation time needed to test multiple batches of antivenoms and yet accurately informed us regarding the binding efficiency of the antivenom product. Venom samples (100 ng) were diluted in a carbonate buffer (pH 9.6) and coated onto 96-well plates. Following the overnight incubation at 4 °C, the unbound venom was washed off using Tris-buffered saline (0.01 M Tris pH 8.5, 0.15 M NaCl) containing 1% Tween 20 (TBST), and incubated with blocking buffer (5% skimmed milk in TBST) for 3 h at room temperature (RT). Following another round of TBST washing, the venom-bound plates were incubated overnight with various dilutions of the antivenom at 4 °C. These antivenom dilutions (1:500, 1:2500, and 1:12500), prepared in a blocking buffer, were added to the plates. Thereafter, unbound antibodies were removed by TBST washing and the plates were incubated at RT for 2 h following the addition of horseradish peroxidase (HRP)-conjugated, rabbit anti-horse secondary antibody (Sigma-Aldrich, St. Louis, MO, USA) diluted at a ratio of 1:1000 in phosphate buffer saline (PBS; pH 7.4). Finally, 100 µL of 2,2/-azino-bis (2-ethylbenzthiazoline-6-sulphonic acid) substrate solution (Sigma-Aldrich, St. Louis, MO, USA) was added, and the resulting optical density was measured at a wavelength of 405 nm for 40 min. The 40th minute was chosen as the endpoint based on the standardisation experiments that showed the highest binding at this time interval.

### 5.9. Immunoblotting

Immunoblotting experiments were carried out using a modified version of the previously reported method [30]. SDS-PAGE (12.5% gel) was used to separate the venoms, which were then transferred onto a nitrocellulose membrane at 25 V and 2.5 A for 7 min according to the manufacturer’s procedure (BioRad, Hercules, CA, USA). After examining the transfer efficiency with Ponceau S reversible stain, the non-specific areas of the membrane were blocked with skimmed milk (5% in TBST) by incubating overnight at 4 °C. The membrane was then washed six times with TBST for 60 min before an overnight incubation at 4 °C with the antivenom at a 1:200 dilution in the blocking buffer. Secondary HRP-conjugated rabbit anti-horse antibody (1:2000 dilution) was added, followed by six TBST washes to remove unbound antivenom. The binding of antivenom to venom was imaged in an iBright CL1000 (Thermo Fisher Scientific, USA) by adding enhanced chemiluminescence substrate, as specified by the manufacturer (Thermo Fisher Scientific, USA).

### 5.10. Immunochromatography

The immunochromatography technique was employed to evaluate the immunoreactivity of conventional Indian antivenoms and the three batches of second-generation antivenoms [12,22]. Here, 100 mg of the polyvalent antivenom was reconstituted in a coupling buffer (200 mM NaHCO_3_ containing 500 mM NaCl, pH 8.5). The column was packed with 3 mL of CNBr activated Sepharose™ 4B matrix (0.25 g/mL) (GE Healthcare, Chicago, IL, USA) and washed with 10 matrix volumes of ice-cold 1 mM HCl, followed by 2 matrix volumes of the coupling buffer. The antivenom affinity matrix was prepared by incubating 60 mg of the polyvalent antivenom (in 2 matrix volumes of coupling buffer) with 3 mL matrix overnight at 4 °C. To prevent non-specific binding, active groups were blocked for 4 h at RT with 6 mL of 100 mM Tris-HCl, pH 8.5. A total of 300 μL matrix of immobilised (5 mg) antivenom was then packed in a column and alternatively washed 6 times with 1 mL of low-pH buffer (100 mM acetate buffer containing 500 mM NaCl, pH 4.0) and 1 mL of high-pH buffer (100 mM Tris-HCl, pH 8.5). The columns were then equilibrated with 5 matrix volumes of binding buffer (20 mM phosphate buffer, 135 mM NaCl, pH 7.4; PBS). Following equilibration, 200 μg of venom dissolved in 150 μL of PBS will be incubated with the matrix for 1 h at 25 °C. The non-retained fractions were collected with 5 matrix volumes of PBS, and the retained fractions were eluted with 5 matrix volumes of 0.1 M glycine-HCl, pH 2.0, and immediately neutralised with 0.1 M Tris-HCl, pH 9.0. The non-retained and retained fractions were concentrated using SpeedVac (Thermo Fisher Scientific, USA) and analysed by reverse-phase HPLC using a C18 column (250 × 4.6 mm, 5 μm particle size, 300 Å pore size) and a DAD detector.

### 5.11. The Intravenous Median Lethal Dose (LD_50_) of Venoms

Prior to the comparative preclinical evaluation of the second-generation antivenom and its conventional counterparts, the toxicities of ‘big four’ snake venoms were evaluated in the murine model. The experimental protocol was approved by the Institutional Animal Ethics Committee (IAEC) (CAF/Ethics/688/2019, Dated 04/04/2019), and was conducted in accordance with the Committee for the Purpose of Control and Supervision of Experiments on Animals (CPCSEA) and WHO guidelines. The potencies of ‘big four’ snake venoms were determined by estimating their LD_50_ values that correspond to the minimum amount of venom required to kill 50% of the test population [26]. Five distinct concentrations of the crude venom were prepared in physiological saline (0.9% NaCl) for injection into the caudal vein of male CD-1 mice. Each venom dose group consisted of 5 CD-1 mice (18 to 22 g) and one control group injected with normal physiological saline. The death and survival patterns for each venom dose group were recorded 24 h post-injection, and Probit analysis was further used to calculate the LD_50_ values [31].

### 5.12. Median Effective Dose (ED_50_) of Antivenoms

The ability of conventional and second-generation antivenoms to neutralise the venom-induced lethal systemic effects was determined using ED_50_ assays. ED_50_ is the minimum amount of antivenom required to protect 50% of the test population injected with the lethal dose of venom [26]. Best performing antivenoms were down-selected based on their in vitro venom binding potential and SE-HPLC and immunochromatographic retention profiles. This selection strategy is optimal from an ethical point of view, as it substantially reduces the number of mice needed for these experiments. Varying amounts of each antivenom were mixed with the challenge dose of venom (5 times the LD_50_) and incubated at 37 °C for 30 min. Each venom-antivenom mixture (*n* = 4) was intravenously injected into a group of male CD-1 mice post-incubation. Survival patterns were recorded 24 h after the injection of the venom-antivenom mixture. ED_50_ values of antivenoms against each venom were then estimated using Probit analyses [31], while the neutralisation potencies of antivenoms were calculated using the equation below [8,32], where ‘n’ represents the number of LD_50_ used as the challenge dose.
$$Antivenom\;neutralisation\;potency\;\left(mg/mL\right)=\frac{\left(n-1\right)\times {\mathrm{LD}}_{50}\;of\;venom\;\left(mg/mouse\right)}{{\mathrm{ED}}_{50}\;\left(mL\right)}$$

Antivenom potencies were also described in terms of the amount of antivenom (in µL) required to neutralise one milligram of venom (Appendix A), following a previously described method [33].

### 5.13. Preclinical Assays

In addition to the lethal systemic effects, snake venoms can also inflict various other pathologies, including haemorrhage, necrosis, myotoxicity, and local paralysis. While the abilities of Indian antivenoms in neutralising systemic effects are relatively better evaluated, their neutralising effects on the local morbid symptoms remain largely uninvestigated. Therefore, we employed the following WHO-recommended murine assays to address this knowledge gap.

The haemorrhagic potential of the crude venom was evaluated by determining the minimum haemorrhagic dose or MHD, which is the minimum amount of venom that causes a 10 mm haemorrhagic lesion under the skin of the test animal (CD-1 mice) [34]. Briefly, five distinct concentrations of the venom diluted in physiological saline (0.9% NaCl) were administered (50 μL/mouse) intradermally into each dose group (*n* = 5). Three hours post-injection, mice were subjected to humane euthanasia by carbon dioxide asphyxiation. Following the dissection of animals, the diameter of haemorrhagic lesions on the underside of the skin was measured with the aid of Vernier calipers, and the mean diameter values were plotted against venom doses to determine the MHD.

Similarly, to estimate the necrotising abilities of venoms, the minimum necrotic dose (MND) was calculated. MND is the minimum amount of venom required to induce the formation of a 5 mm necrotic lesion on the underside of the skin of the test animal. The size of the necrotic lesions was measured after 72 h of venom injection and the mean diameter of the lesions was plotted against venom concentrations [26].

### 5.14. Morbidity Neutralisation Tests

The second-generation and conventional antivenoms were analysed for their ability to neutralise venom-induced haemorrhage by estimating their respective MHD-median effective doses (MHD_50_) or the minimum volume of antivenom (in µL) necessary to reduce the diameter of haemorrhagic lesions by 50% when injected with the challenge dose of venom (5X the MHD). The challenge dose was incubated with five distinct dilutions of antivenom at 37 °C for 30 min, and 50 μL of this mixture was injected intradermally in each dose group of CD-1 mice. Mice were asphyxiated with carbon dioxide three hours post-injection, dissected, and the diameter of haemorrhagic lesions on the inner side of the skin was measured [26].

To determine the necrosis neutralising potency, a venom concentration corresponding to two MND was used as the challenge dose and a similar procedure as MHD_50_ was followed. The dissection was performed on the third-day post the injection of the venom-antivenom mixture, and the minimum amount of antivenom (in µL) able to reduce the diameter of the necrotic lesion by 50% compared to the control was determined as the MND_50_ of the antivenom [26].

### 5.15. Statistical Analyses

Two-way ANOVA and unpaired t-tests were used to evaluate statistical differences in the ELISA and preclinical assay results. These comparisons were conducted using GraphPad Prism (GraphPad Software 9.0, San Diego, CA, USA, www.graphpad.com, accessed on 7 December 2021).

## Figures and Tables

**Figure 1 toxins-14-00168-f001:**
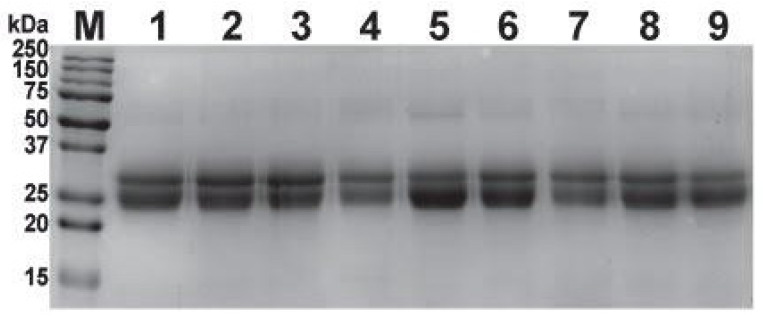
This figure depicts the SDS-PAGE profiles of the three batches of second-generation antivenom and conventional Indian antivenoms. M: protein marker; 1: SIIPL-01; 2: SIIPL-02; 3: SIIPL-03; 4: Premium Serums And Vaccines Pvt. Ltd.; 5: VINS BioProducts Ltd.; 6: Bharat Serums; 7: Haffkine Pharmaceuticals; 8: Virchow; 9: Biological E. Ltd.

**Figure 2 toxins-14-00168-f002:**
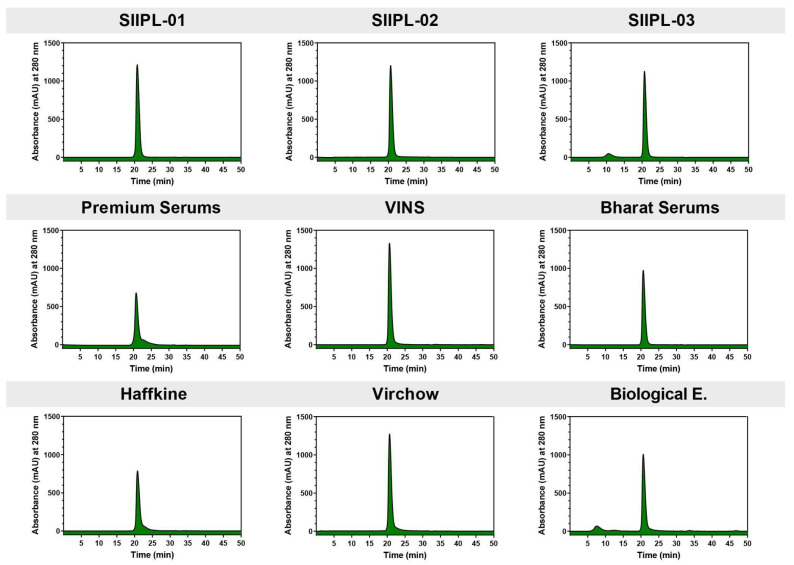
SE-HPLC profiles of the improved second-generation and conventional antivenoms.

**Figure 3 toxins-14-00168-f003:**
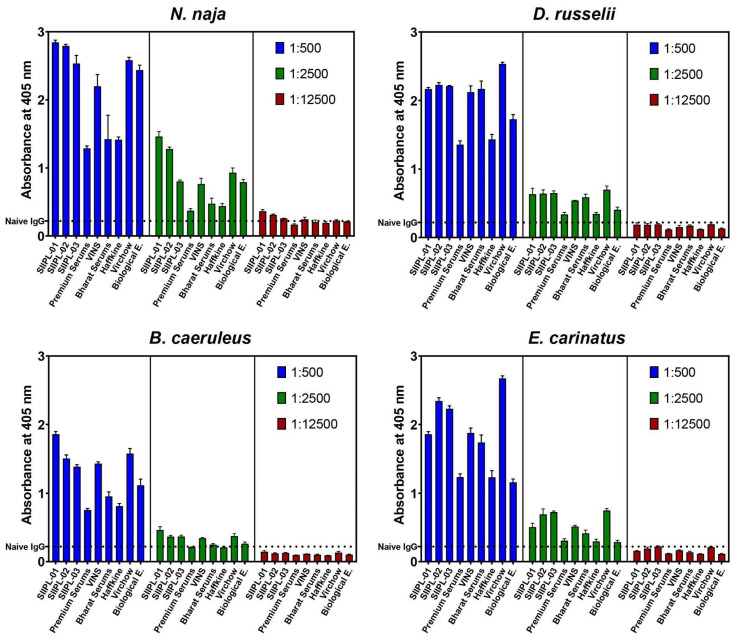
This figure shows the binding efficiency of improved second-generation and conventional antivenoms against the ‘big four’ snake venoms.

**Figure 4 toxins-14-00168-f004:**
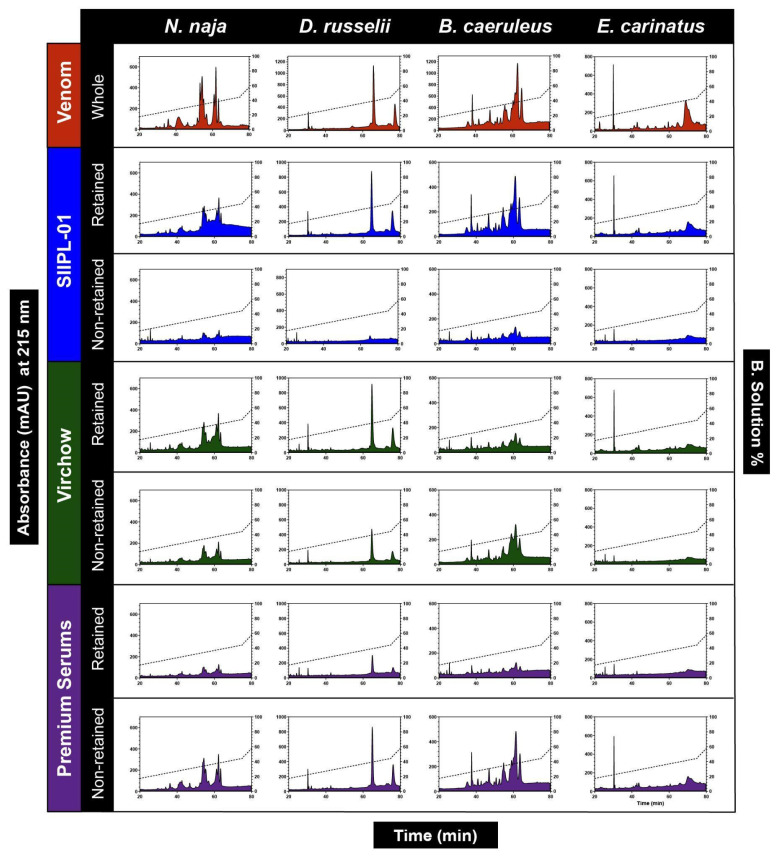
Immunochromatography profiles of the second-generation (SIIPL-01) and conventional (Virchow and Premium Serums) antivenoms against the ‘big four’ snake venoms.

**Figure 5 toxins-14-00168-f005:**
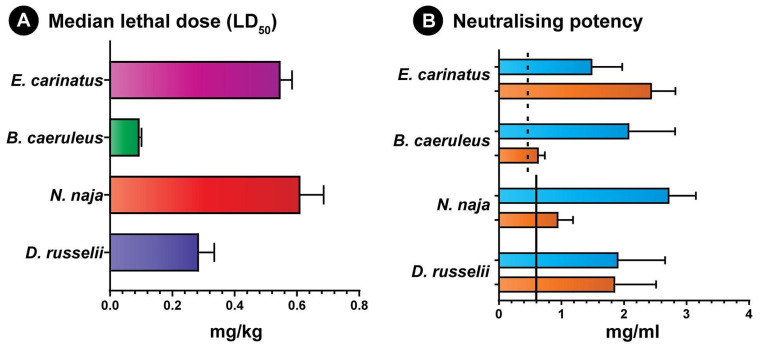
(**A**) Venom potencies of the ‘big four’ snakes from Tamil Nadu and (**B**) the neutralisation potencies (the amount of venom neutralised by 1 mL of the antivenom) of the second-generation (SIIPL-01; in blue) and Virchow (in orange) antivenoms. In panel B, the currently marketed potency of conventional products against *N. naja* and *D. russelii* venoms is denoted by a solid line (0.60 mg/mL), while the marketed neutralisation potency against *B. caeruleus* and *E. carinatus* venoms is shown as a dotted line (0.45 mg/mL).

**Figure 6 toxins-14-00168-f006:**
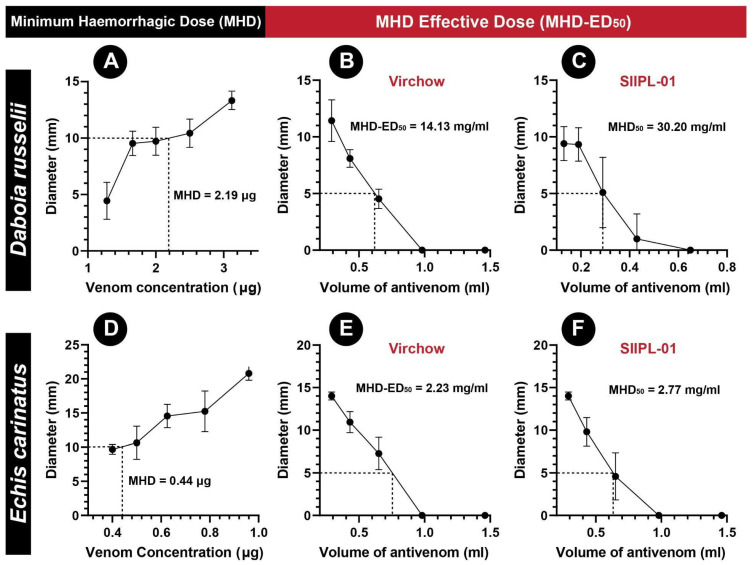
The MHD of venoms and MHD-ED_50_ of antivenoms. (**A**) MHD of *D. russelii* venom, (**B**) MHD-ED_50_ of Virchow antivenom against *D. russelii* venom, (**C**) MHD-ED_50_ of SIIPL antivenom against *D. russelii* venom, (**D**) MHD of *E. carinatus* venom, (**E**) MHD-ED_50_ of Virchow antivenom against *E. carinatus* venom, and (**F**) MHD-ED_50_ of SIIPL antivenom against *E. carinatus* venom.

**Table 1 toxins-14-00168-t001:** Physicochemical properties of commercial Indian antivenoms.

Manufacturer	Batch No.	Manufacturing (M) and Expiry (E) Dates	Formulation	Protein Content (mg/mL)	Appearance	Colour after Reconstitution	Odour	pH	Reconstitution Time
Second-generation’ antivenom									
Serum Institute of India (SIIPL-01)	SIIPL-01	NA	F(ab’)_2_	39.78 ± 0.98	Amorphous powder	Opaque	Odourless	5.74	50 s
Serum Institute of India (SIIPL-02)	SIIPL-02	NA	F(ab’)_2_	29.33 ± 7.87	Amorphous powder	Clear colourless	Weak	6.13	1.10 min
Serum Institute of India (SIIPL-03)	SIIPL-03	NA	F(ab’)_2_	14.29 ± 0.88	Amorphous powder	Clear colourless	Weak	5.80	30 s
Conventional Indian antivenoms									
Premium Serums & Vaccines Pvt. Ltd. (Premium Serum)	ASVS(1)Ly-015	NA	F(ab’)_2_	16.79 ± 4.26	Amorphous powder	Slight yellow	Strong	6.25	2.46 min
VINS Bioproducts Ltd. (VINS)	01AS18067	M: 11/2018 E: 10/2022	F(ab’)_2_	19.34 ± 7.38	Amorphous powder	Clear colourless	Weak	6.49	5.23 min
Bharat Serums and Vaccines Ltd. (Bharat Serums)	A05318020	M: 01/2018 E: 12/2021	F(ab’)_2_	29.63 ± 10.8	Amorphous powder	Opaque	Strong	6.37	44 s
Haffkine BioPharmaceutical Corporation Ltd. (Haffkine)	AS180611	M: 06/2018 E: 11/2022	F(ab’)_2_	30.50 ± 5.97	Amorphous powder	Pale yellow	Weak	6.40	1.17 min
Virchow Biotech Private Ltd. (Virchow)	PAS00718	M: 09/2018 E: 08/2022	F(ab’)_2_	19.54 ± 3.86	Amorphous powder	Clear colourless	Weak	6.63	2.10 min
Biological E. Ltd. (Biological E.)	BAS00218	NA	F(ab’)_2_	34.48 ± 6.71	Amorphous powder	Clear colourless	Weak	6.19	1.32 min

This table outlines the ‘second-generation’ and commercial Indian antivenoms investigated in this study, with corresponding batch numbers; manufacturing and expiration dates; and protein concentrations of the vials, as estimated by Bradford method. NA: not available.

## Data Availability

The raw proteomics data generated for this study can be found at PRIDE Database (Accession No: PXD029436).

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
