# Peer review of "The Preclinical Evaluation of a Second-Generation Antivenom for Treating Snake Envenoming in India"

_toxins, 2022, doi:10.3390/toxins14030168_

Round 1

Reviewer 1 Report

After reading and rereading the manuscript, there are a few key points that I consider should be addressed:

  • In the Introduction or Discussion section, a new paragraph could be introduced, to reflect on the reasons as to why purification methods have never before been used to enhance the antivenom effectiveness. The methods described in the manuscript are not extremely difficult, and the possible advantages of purification are significant. Why did no research group try to purify antivenoms until now?
  • Table 1 includes the SIIPL antivenom. Please rephrase the table description to reflect that both commercial and second-generation antivenoms are presented. As the Toxins format requires the Results before the Methods section, it is unclear at this point in the manuscript that SIIPL represents the purified antivenoms. I would recommend some explanation or a table footer, to explain what SIIPL is at this point of the manuscript.
  • Table 1: There are some differences in the physicochemical properties of the SIIPL antivenoms. Could the authors provide some explanation as to why these differences occur?
  • Regarding SDS-PAGE and SE-HPLC: Some explanation should be given in the Results section about what impurities could have been detected by SE-HPLC. Some of them are mentioned in the Methods section, which, due to the template, is at the end of the manuscript.
  • Lines 99-108: There are some formatting problems in this paragraph, please verify and correct.
  • Regarding MS analysis: Why not evaluate all antivenoms for a complete comparison?
  • Regarding Chromatographic purification: I might have overlooked the information, but there seems to be no clear indication as to what exactly has been subjected to purification to obtain the second-generation antivenoms. Where the commercially available antivenoms purified? If so, which ones?
  • I would recommend more emphasis on the purification procedure and results. It is the starting point for the entire experiment, yet it is described in a single paragraph in the Methods section. From an analytical point of view, it is at least as important as the subsequent tests performed with the obtained fractions.
  • Based on the Methods section, it is not clear how many duplicates/measurements have been performed for each sample (second-generation or conventional antivenom). This is a relevant information, as a single measurement, for a single production lot of a sample does not provide adequate statistical significance to the results. Kindly state in the method description the number of duplicates used. If no duplicates where used, the results should be validated by further experiments prior to publication.

I would like to congratulate the authors for the exceptionally well-written manuscript. If the results are supported by enough number of duplicates, then a new direction in antivenom production could finally open, to help address the severe problems caused by conventional antivenom therapy .

Author Response

We are thankful to the reviewer for their kind words of appreciation. Their suggestions helped in improving the overall quality of the manuscript.

Comments and Suggestions for authors

After reading and rereading the manuscript, there are a few key points that I consider should be addressed:

  • In the Introduction or Discussion section, a new paragraph could be introduced, to reflect on the reasons as to why purification methods have never before been used to enhance the antivenom effectiveness. The methods described in the manuscript are not extremely difficult, and the possible advantages of purification are significant. Why did no research group try to purify antivenoms until now?

We thank the reviewer for the suggestion. We have now included a paragraph stating the reason why purification has not been used before in the antivenom manufacturing process in the discussions (Line number 232-239).

  • Table 1 includes the SIIPL antivenom. Please rephrase the table description to reflect that both commercial and second-generation antivenoms are presented. As the Toxins format requires the Results before the Methods section, it is unclear at this point in the manuscript that SIIPL represents the purified antivenoms. I would recommend some explanation or a table footer, to explain what SIIPL is at this point of the manuscript.

Thanks very much for this suggestion. We have now revised Table 1 to provide clarity (Line number 69-71).

  • Table 1: There are some differences in the physicochemical properties of the SIIPL antivenoms. Could the authors provide some explanation as to why these differences occur?

The production of second generation antivenoms was a pilot scale R&D initiative. During the manufacturing process, hyperimmune plasma was harvested from several equines. Since the equine plasma is an animal derived product, it often leads to compositional differences in the test batches. These processes will be optimized and validated for the commercial production of the ‘second generation’ antivenoms to minimize batch to batch variation in the finished product. 

  • Regarding SDS-PAGE and SE-HPLC: Some explanation should be given in the Results section about what impurities could have been detected by SE-HPLC. Some of them are mentioned in the Methods section, which, due to the template, is at the end of the manuscript.

We have now listed impurities present in the antivenoms in the SE-HPLC section (Line number 89-91).

  • Lines 99-108: There are some formatting problems in this paragraph, please verify and correct.

Thanks for pointing out this formatting issue that appeared post-submission formatting of the manuscript. We have now reformatted the paragraph (Line number 109-119).

  • Regarding MS analysis: Why not evaluate all antivenoms for a complete comparison?

We have selected three antivenoms (SIIPL-01, Virchow and Premium Serums) for the Mass spectrometry and immunochromatography based on the outcomes of the in vitro binding experiments. This streamlined strategy was followed during the COVID lockdowns given the strategic and financial constraints.

  • Regarding Chromatographic purification: I might have overlooked the information, but there seems to be no clear indication as to what exactly has been subjected to purification to obtain the second-generation antivenoms. Where the commercially available antivenoms purified? If so, which ones?

Detailed method for production and chromatographic purification to obtain second-generation antivenom has now been added in the methods section (Line number 339-377).

  • I would recommend more emphasis on the purification procedure and results. It is the starting point for the entire experiment, yet it is described in a single paragraph in the Methods section. From an analytical point of view, it is at least as important as the subsequent tests performed with the obtained fractions.

Detailed method for chromatographic purification has now been added in the methods section (Line number 339-377).

  • Based on the Methods section, it is not clear how many duplicates/measurements have been performed for each sample (second-generation or conventional antivenom). This is a relevant information, as a single measurement, for a single production lot of a sample does not provide adequate statistical significance to the results. Kindly state in the method description the number of duplicates used. If no duplicates where used, the results should be validated by further experiments prior to publication.

While we completely agree that the results are not statistically validated, such rigorous comparisons are seldom executed in the literature for venom neutralisation studies. Such comparisons are not undertaken given the financial and ethical constraints. Please note that the three antivenom bulk batches were prepared independently in this process by immunising a large number of horses, which itself is a very costly affair. The manufacturer kept aside these horses for this pilot experiment as the resultant antivenoms could not be used for commercial purposes. Obtaining any additional test batches for pilot experiments would be financially constraining to antivenom manufacturers. Considering this, we have not undertaken statistical comparisons in this preclinical study. We assure the reviewer that rigorous statistical tests would be conducted and these antivenom would be clinically validated in the very near future. This publication would provide the needed evidence to demonstrate the power and need for simpler steps to enhance the quality of conventional antivenom products in India.

I would like to congratulate the authors for the exceptionally well-written manuscript. If the results are supported by enough number of duplicates, then a new direction in antivenom production could finally open, to help address the severe problems caused by conventional antivenom therapy.

We are thankful to the reviewer for their kind words of appreciation. Their suggestions helped in improving the overall quality of the manuscript.

Reviewer 2 Report

Authors reported the feasibility and effectiveness of a simple pre-processing steps that greatly enhanced the dose-effectiveness and neutralisation potential of conventional antivenom, which they called “second-generation” antivenoms, using chromatographic purification method. The key contributing factor to the success appears to be enhanced purity, and they have demonstrated the improved efficacy through preclinical assays against toxic effects of the big four Indian snakes. The study is interesting and the finding is meaningful for the field of snakebite treatment. Potentially antivenom producers will benefit from this study through which the efficacy of their products can be improved. The work is publishable and is relevant for the journal TOXINS. There are, however, some comments and suggestions from the reviewer for improvement of the manuscript. It is hoped that the existing data can be discussed in greater extents, some method details can be provided, and the paper be more inclusive of references or credits from previous relevant studies.

Comments/Suggestions:

Table 1. Footnote - provide definition for NA.

Line 69-70: Electrophoresis revealed the presence of two bands between 37 and 25 kDa (Figure 1), corroborating with the previously reported profiles of F(ab’)2 preparations from equine origin [15-18]. --The range of MW should be noted as between 25 to (below) 37 kDa. Similar banding patterns within the range have been shown in a number of studies on horse antivenom products in Asia but none has been referenced to. These studies suggested the bands to represent the heavy and the light chains, respectively of F(ab)'2 under reducing conditions. Suggest authors to refer and include the discussion of the bands and their possible identities.

References:

Indonesian antivenom: https://www.nature.com/articles/srep37299

Thai, Chinese antivenoms: https://www.sciencedirect.com/science/article/pii/S0001706X19315116

line 80-85: These Size-exclusion chromatograms can perhaps provide information about the MW of eluted proteins with the corresponding peaks. Were the peaks monitored and calibrated with molecular weight standards? Please provide the information that can help identify the proteins or their molecular weights, for proteins in the main peaks. In addition, please explain how or why the peak tailing was claimed to be impurities, and please suggest what the minor peak at 10 min RT for SIIPL-03 and Biological E product could be.

Figure 5 legend: pls include the defintiion of neutralising potency in this study.

Line 263-264 “Recent studies have unravelled the inefficaciousness of Indian antivenoms in countering pathologies inflicted by the disparate populations of ‘big four’ snakes and their close relatives [3,4,6-9,11-13].” --- Indeed, the Big Four populations are disparately distributed (within and beyond India) and close relatives need to be considered in antivenom treatment. Findings which have been reported for Big Four and close relatives from various or other geographical localities with regard to their venom neutralization (with limited efficacy) using Indian antivenoms may have been included by authors, and perhaps to discuss in slightly more details (antivenom efficacy variation associated with geographical factor and the impact on use in foreign countries). Please see the following related works for referencing:

  1. Bungarus sindanus, Pakistan, using Indian antivenom - https://www.sciencedirect.com/science/article/pii/S1874391918303890
  2. Bungarus caeruleus, Sri Lanka, India, Pakistan, Indian antivenom - https://www.sciencedirect.com/science/article/pii/S1874391917301446?via%3Dihub
  3. Naja naja, Pakistan, Indian antivenom included - https://pubmed.ncbi.nlm.nih.gov/27022154/

Line 316: Regarding "supernatant..." is it possible to briefly include how this (supernatant) came from – perhaps, briefly state the immunization, blood and serum collection, antibody collection processes etc.?

Author Response

Comments and Suggestions for Authors

Authors reported the feasibility and effectiveness of a simple pre-processing steps that greatly enhanced the dose-effectiveness and neutralisation potential of conventional antivenom, which they called “second-generation” antivenoms, using chromatographic purification method. The key contributing factor to the success appears to be enhanced purity, and they have demonstrated the improved efficacy through preclinical assays against toxic effects of the big four Indian snakes. The study is interesting and the finding is meaningful for the field of snakebite treatment. Potentially antivenom producers will benefit from this study through which the efficacy of their products can be improved. The work is publishable and is relevant for the journal TOXINS. There are, however, some comments and suggestions from the reviewer for improvement of the manuscript. It is hoped that the existing data can be discussed in greater extents, some method details can be provided, and the paper be more inclusive of references or credits from previous relevant studies.

We are grateful to the reviewers for their helpful suggestions and the words of appreciation.

Comments/Suggestions:

  • Table 1. Footnote - provide definition for NA.

We have now added the definition of NA in table legends (Line number 71).

  • Line 69-70: Electrophoresis revealed the presence of two bands between 37 and 25 kDa (Figure 1), corroborating with the previously reported profiles of F(ab’)2 preparations from equine origin [15-18]. --The range of MW should be noted as between 25 to (below) 37 kDa. Similar banding patterns within the range have been shown in a number of studies on horse antivenom products in Asia but none has been referenced to. These studies suggested the bands to represent the heavy and the light chains, respectively of F(ab)'2 under reducing conditions. Suggest authors to refer and include the discussion of the bands and their possible identities.

References:

Indonesian antivenom: https://www.nature.com/articles/srep37299

Thai, Chinese antivenoms: https://www.sciencedirect.com/science/article/pii/S0001706X19315116

We have now made this correction. We have also added references to support this statement (Line number 76-78).

  • line 80-85: These Size-exclusion chromatograms can perhaps provide information about the MW of eluted proteins with the corresponding peaks. Were the peaks monitored and calibrated with molecular weight standards? Please provide the information that can help identify the proteins or their molecular weights, for proteins in the main peaks. In addition, please explain how or why the peak tailing was claimed to be impurities, and please suggest what the minor peak at 10 min RT for SIIPL-03 and Biological E product could be.

We employed SEC to evaluate the overall purity and quality of the product; that is, to evaluate the presence of dimers or aggregates. To identify the components in the major peaks, we have employed mass spectrometry. The mass spectrometric analysis revealed that IgGs constituted the majority of the protein content, while impurities (e.g., albumin, alpha-macroglobulin, fibrinogen, fibronectin, haptoglobin and plasminogen prothrombin and serpin) formed a minor fraction of all three antivenoms (Line number 167-169).

The peak tailing has been attributed to the presence of impurities (e.g., albumin, alpha-macroglobulin, fibrinogen, fibronectin, haptoglobin, plasminogen, prothrombin, etc.) along with IgGs (Tan CH, 2016). The minor peak at 10 mins in SEC profiles of the SIIPL and Biological E antivenoms are dimers of F(ab)'2 (Tan CH, 2016), this is now mentioned in line number 93.

  • Figure 5 legend: pls include the defintiion of neutralising potency in this study.

We have now included the definition of neutralisation potency in the legend of Figure 5 (Line number 189-190).

  • Line 263-264 “Recent studies have unravelled the inefficaciousness of Indian antivenoms in countering pathologies inflicted by the disparate populations of ‘big four’ snakes and their close relatives [3,4,6-9,11-13].” --- Indeed, the Big Four populations are disparately distributed (within and beyond India) and close relatives need to be considered in antivenom treatment. Findings which have been reported for Big Four and close relatives from various or other geographical localities with regard to their venom neutralization (with limited efficacy) using Indian antivenoms may have been included by authors, and perhaps to discuss in slightly more details (antivenom efficacy variation associated with geographical factor and the impact on use in foreign countries). Please see the following related works for referencing:
  1. Bungarus sindanus, Pakistan, using Indian antivenom - https://www.sciencedirect.com/science/article/pii/S1874391918303890
  2. Bungarus caeruleus, Sri Lanka, India, Pakistan, Indian antivenom - https://www.sciencedirect.com/science/article/pii/S1874391917301446?via%3Dihub
  3. Naja naja, Pakistan, Indian antivenom included - https://pubmed.ncbi.nlm.nih.gov/27022154/

We have now included a line on this in the discussion section (Line number 299-302).

  • Line 316: Regarding "supernatant..." is it possible to briefly include how this (supernatant) came from – perhaps, briefly state the immunization, blood and serum collection, antibody collection processes etc.?

The detailed method for the production of second-generation antivenom has now been added in the methods section (Line number 339-377).

Reviewer 3 Report

The manuscript “The preclinical evaluation of a second-generation antivenom for treating snake envenoming in India” examines how improvements could be made to existing antivenoms to reduce snakebite mortality and morbidity in Indian. This is very important work, as the development of entirely new antivenoms, such as the use of monoclonal antibodies, is many years from being available in a clinical setting to treat snakebite. This manuscript is well-written with nice figures, and the experiments are well done. I do have a few suggestions that would improvement the manuscript quality:

  1. I would add to the introduction more information on the complete process to produce current antivenoms – such as if caprylic acid is used or IgG affinity purification (protein A or G)? What is the difference between the manufacturers in antivenom preparations? This background will clarify the improvements being made.

  1. To follow my first suggestion, there should be more information on how the improvements are being carried out. What was done to produce SIIPL-01, SIIPL-02, and SIIPL-03? Is there a different process between them? The methods are a bit vague.

  1. What would be the cost of adding this secondary chromatographic purification step to Indian antivenom preparation? Is this feasible for antivenom production?

  1. Section 5.12-5.13: I assume university ethical approval was obtained to perform the animal experiments? Could you please provide these details, such as the approved protocol number or project license?

Minor comments:

Line 21, and Line 53: Is this simpler processing? An additional step would be being added to antivenom preparation…

Table 1: What venoms are used to make these antivenoms? This information could be added to the table, or in the text. It could also explain poor performing antivenoms and should be discussed in this context.

Section 2.4: Text formatting is variable in this section and should be made consistent.

Section 2.6: Could you quantify the abundance of IgGs and contaminants? This would provide a more robust result.

Line 192: Figure 6A & D? This might be a typo?

Line 319: What resin was used for the chromatography column? Protein G? Is the resin used commonly available or custom?

Author Response

The manuscript “The preclinical evaluation of a second-generation antivenom for treating snake envenoming in India” examines how improvements could be made to existing antivenoms to reduce snakebite mortality and morbidity in Indian. This is very important work, as the development of entirely new antivenoms, such as the use of monoclonal antibodies, is many years from being available in a clinical setting to treat snakebite. This manuscript is well-written with nice figures, and the experiments are well done. I do have a few suggestions that would improvement the manuscript quality:

1. I would add to the introduction more information on the complete process to produce current antivenoms – such as if caprylic acid is used or IgG affinity purification (protein A or G)? What is the difference between the manufacturers in antivenom preparations? This background will clarify the improvements being made.

Thanks for suggesting this. We have now explained the complete process of antivenom production in the methods section (line number 339-377) and a paragraph in the discussion section has also (Line number 232-239).

2. To follow my first suggestion, there should be more information on how the improvements are being carried out. What was done to produce SIIPL-01, SIIPL-02, and SIIPL-03? Is there a different process between them? The methods are a bit vague.

The detailed method for the production of second-generation antivenom has now been added in the methods section (Line number 339-377). SIIPL-01, SIIPL-02, and SIIPL-03 are three independently manufactured batches of the second-generation antivenom.

3. What would be the cost of adding this secondary chromatographic purification step to Indian antivenom preparation? Is this feasible for antivenom production?

The present study was a pilot scale experiment, wherein commercially available ion-exchange resin (Bio-Rad Laboratories, USA) was used for the purification of the antivenom product. Incorporation of this purification strategy for the commercial manufacture of antivenoms may require a decent capital investment, but would still be affordable to the majority of antivenom manufacturers.

4. Section 5.12-5.13: I assume university ethical approval was obtained to perform the animal experiments? Could you please provide these details, such as the approved protocol number or project license?

These details (clearance numbers, etc.) are included in the manuscript and were verified by the editorial staff. But they have excluded these details to facilitate the double-blind review process.

Minor comments:

  • Line 21, and Line 53: Is this simpler processing? An additional step would be being added to antivenom preparation…

These steps are financially and strategically much simpler for the Indian antivenom manufacturers to incorporate, instead of having to invest in facilities to express recombinant antibodies for snakebite treatment - something that seems near impossible for the majority of Indian antivenom manufacturers.

  • Table 1: What venoms are used to make these antivenoms? This information could be added to the table, or in the text. It could also explain poor performing antivenoms and should be discussed in this context.

All antivenoms used in this study were manufactured using the venoms of the spectacled cobra (Naja naja), Russell’s viper (Daboia russelii), common krait (Bungarus caeruleus) and saw-scaled viper (Echis carinatus) that were supplied by the Irula or Haffkine Institute. A line has been added in the results section (Line number 61-63).

  • Section 2.4: Text formatting is variable in this section and should be made consistent.

We have reformatted the text in this section 2.4 and made it consistent (Line number 109-119).

  • Section 2.6: Could you quantify the abundance of IgGs and contaminants? This would provide a more robust result.

The relative quantification of components present in antivenoms was not determined in this study as these experiments were conducted during the COVID lockdown. We agree that this would be an interesting data, which we hope to generate in the near future.

  • Line 192: Figure 6A & D? This might be a typo?

We thank the reviewer for pointing out the typographical error. We have added the correct citation in the revised version of the manuscript.

  • Line 319: What resin was used for the chromatography column? Protein G? Is the resin used commonly available or custom?

We have used commercially available ion exchange resin (Bio-Rad Laboratories, USA) for chromatographic purification of antivenom and this is now mentioned in the manuscript.

Round 2

Reviewer 1 Report

Thank you to the authors for considering the suggestions and revisions received. All of them are acceptable.

There is a single point I would like to recommend further:

Author's response: While we completely agree that the results are not statistically validated, such rigorous comparisons are seldom executed in the literature for venom neutralisation studies. Such comparisons are not undertaken given the financial and ethical constraints. Please note that the three antivenom bulk batches were prepared independently in this process by immunising a large number of horses, which itself is a very costly affair. The manufacturer kept aside these horses for this pilot experiment as the resultant antivenoms could not be used for commercial purposes. Obtaining any additional test batches for pilot experiments would be financially constraining to antivenom manufacturers. Considering this, we have not undertaken statistical comparisons in this preclinical study. We assure the reviewer that rigorous statistical tests would be conducted and these antivenom would be clinically validated in the very near future. This publication would provide the needed evidence to demonstrate the power and need for simpler steps to enhance the quality of conventional antivenom products in India.

Further recommendation: Indeed, I believe the preliminary results should be published to promote this new direction of research. However, a "Limitations of the study" section should be included, with the explanations from above (even if in a shorter form).

Author Response

We thank the reviewer for the suggestion. We have now added the ‘Limitations of the study’ sub-section (3.5) at the end of the Discussion section (Line number 308-313).